# How frames and narratives in press releases shape newspaper science articles: the case of ocean plastic pollution.

Aike Vonk<sup>1</sup>, Mark Bos<sup>1</sup>, Erik van Sebille<sup>1,2</sup>

<sup>1</sup>Public Engagement and Science Communication, Freudenthal Institute, Utrecht University, 3584CC Utrecht, The Netherlands <sup>2</sup>Physical Oceanography, Marine and Atmospheric Research, Utrecht University, 3584CC Utrecht, The Netherlands

Correspondence to: Aike Vonk (a.n.vonk@uu.nl)

10

Abstract: Although framing of scientific topics in the media has been widely studied, relatively little is known about the origins of these frames. Since (geo)science journalism often relies on university press releases, this study investigates how peer-reviewed research on ocean plastic is framed in university press releases and newspaper articles. Using Entman's framing theory, we examine how ocean plastic science is described through problem definitions, causes, moral judgments and solutions. Additionally, we studied narrative elements like personalization, dramatisation, and emotionalisation. Using a novel visualisation technique, we combine quantitative and qualitative analysis to reveal shifts in content and show which information is added, adapted or omitted by journalists when covering the research in the newspaper. Our results show that journalists often adopt framing and quotes directly from press releases, with scientists consistently portrayed as central figures, either as heroes addressing the plastic crisis or as warners highlighting its dangers. Although some articles add additional context, especially in assigning responsibility, the social embedding of the problem remains limited, resulting in personalization of the science instead of ocean plastic pollution. Information in the press release is almost never validated by a scientist not involved in the study. Moreover, non-scientific actors are rarely brought up and perspectives of victims or causers are often missing. These findings demonstrate that press releases strongly shape how ocean plastic research is framed in the media, offering research institutes an opportunity to promote more socially contextualised and relevant ocean science communication.

**Short summary:** Research institutes communicate scientific findings through press releases, which journalists use to write news articles. We examined how journalists use content from press releases about ocean plastic research. Our findings show that they closely follow the press releases story, primarily quoting involved scientists without seeking external perspectives. Causing the focus to stay on researchers, personalizing science rather than addressing the broader societal dimensions of plastic pollution.

#### 1. Introduction

The media play an important role in shaping public understanding on ocean issues. They are a major source of information about marine science and therefore influence how risks are perceived (Kramm et al., 2022). Although public awareness of ocean issues is increasing, perceptions of the most pressing problems often differ from those of scientists. For example, while the public often identify ocean pollution as their greatest concern, scientists are more likely to highlight climate change as the most urgent problem (Lotze et al., 2018). Scientists suggest that the disproportionate media coverage of ocean plastic, compared to other areas of ocean science (Pinto et al., 2020; Thompson-Saud et al., 2018), may have contributed to this gap in perception (Tiller et al., 2019).

How research gets in the news is increasingly determined by press releases from communication professionals, especially as the number of dedicated science journalists decreases and universities expand their media activities (Autzen, 2014; Peters et al., 2008; Comfort et al., 2022; Vögler & Schäfer, 2020). As a result, press releases have become an important source of information for journalists (Schafraad & Van Zoonen, 2020), who often draw directly from them when writing newspaper articles (Nelissen & Hendrickx, 2023; Van Leuven et al., 2015). These press releases do more than summarize research findings, they shape the tone and framing of problems like ocean plastic pollution (Vonk et al., 2024a). In some cases, parts of these press releases are copied verbatim (Comfort et al., 2022), a practice known as "copy-paste journalism" or "churnalism" (Autzen, 2014; Kroon & Schafraad, 2013; Van Leuven et al., 2015). As a result, the way research institutes frame science, plays an important role in how research is presented in newspaper articles.

In this study, we examine how scientific knowledge about ocean plastic spreads from research institutions to the public, focusing on the role of press releases in shaping media messages. Ocean plastic research forms a suitable case study because it is highly visible in both scientific communication and public discourse. While multiple studies show how ocean plastic is framed in public media, less attention has been given to the origins of these frames. Our study fills this gap, and explores contextual shifts among press releases and subsequent newspaper articles, by examining changes in framing and narrative strategies. By comparing how framing is used in both press release and newspaper article, we can better understand the role of the press release in frame construction. Furthermore, analysis of narratives, actors and quotes provides information about the social context in which ocean plastic research is placed and the people deemed important in the conversation about ocean plastic pollution.

# 55 1.1 Communication challenges and the role of (science) journalism

Understanding the impacts of ocean problems, like ocean plastic, can be challenging for people, as ocean problems are deeply interconnected and influenced by multiple stressors, making it difficult to isolate and fully understand the impact of a single issue (Kelly et al., 2022). Moreover, ocean science often requires the use of complex biological, chemical and physical

methods. Understanding these methods can be complicated as they require a relatively large amount of prior knowledge to comprehend it properly. Additionally, the ocean, and particularly the deep sea, is largely invisible and physically remote, contributing to a sense of detachment and making it harder for the public to perceive its relevance to society (Schuldt et al., 2016).

Effectively communicating ocean science to the general public therefore requires translating complex research into content that is accessible and meaningful for newspaper readers. In doing so, journalists encounter a number of challenges. In several European countries, journalists report that ocean science receives limited attention in newspapers, partly due to understaffed newsrooms and a shortage of specialized science reporters. Additionally, the complexity and technical nature of ocean science makes it difficult to accurately interpret research findings. Journalists often emphasize the need to consult directly with scientists to clarify and verify information, but such access is not always feasible. Time pressures further complicate their work, leaving journalists with little opportunity for in-depth investigation. As a result, they tend to rely more heavily on easily accessible international sources, often copying content related to ocean problems (Pinto & Matias, 2023).

One of the tasks of science journalists is to interpret the research findings from press releases and peer-reviewed articles, and present them in an accessible way to a wide audience (Autzen, 2014; Wormer, 2008; Korthagen, 2016). In line whith this, it is said that to communicate ocean science effectively, communication must not only present scientific facts, but also contextualise them within broader social and cultural frameworks (United Nations Educational, Scientific and Cultural Organization, 2021). Because by embedding ocean science in society, science journalists make complex issues more understandable and promote greater public engagement and pro-environmental behaviour towards the ocean (Catalano et al., 2019; Stoll-Kleemann, 2019).

To make research relevant and meaningful to their audiences, it is expected that journalists modify press release text, especially when embedding the research within local or cultural narratives. For example, press releases distributed via platforms like EurekAlert! are picked up by news outlets in multiple countries (Vonk et al., 2024b). When covering such international research, journalists often localise the content to enhance cultural resonance (Bassnett, 2005). In addition, journalists may choose to include voices from other sectors, such as government or public health officials, to broaden the perspective described in the press release and add societal relevance to the research (Sharp et al., 2021), as scientific press releases on ocean plastic typically only include quotes from the scientists involved in the study (Vonk et al., 2024a). A selective use of quotes results in a different contextualisation of the research in news articles compared to the original press release (Sharp et al., 2021).

#### 1.2 Telling the story of ocean plastic pollution

In communicating research, journalists make choices about what information they highlight or omit, thereby determining how science is framed in the public arena (Nisbet & Mooney, 2007; Yang & Hobbs, 2020). These frames help structure complex

information and make it meaningful (Nisbet, 2009), thereby guiding the interpretation of a story (Druckman & Lupia, 2017). Narratives can further support this process by placing scientific research within a human context and linking it to everyday life (Dahlstrom, 2014). They help make abstract environmental problems, such as ocean plastic, more understandable and relevant to the public (Cooper & Nisbet, 2016). Given that frames and narratives strongly shape how audiences interpret scientific information, it is important to understand how they are used to communicate ocean plastic research.

Newspaper coverage about ocean plastic often emphasizes risks, damages, and the negative consequences of ocean plastic pollution, while opportunities or benefits related to plastic use are rarely addressed (Welzenbach-Vogel et al., 2022). In addition, ocean plastic pollution is typically portrayed as a threat to marine ecosystems rather than to human health, which may reinforce the perception that ocean plastic is a distant problem with little relevance to people's everyday lives (Henderson & Green, 2020). Moreover, news coverage of microplastics focuses on risks and scientific knowledge, placing responsibility for risk reduction mostly on consumers and policymakers, while the role of industry is hardly mentioned (Schönbauer & Müller, 2021).

#### 2 Theoretical framework

# 2.1 The influence of scientific press releases in frame-building

Research on framing often identifies predetermined frames or looks at specific textual elements, for example positive or negative framings regarding plastic (Welzenbach-Vogel et al., 2022) or the actors deemed important in the story (Schönbauer & Müller, 2021). While these approaches provide valuable insights, they risk confirming expectations by narrowing the scope of analysis to what researchers already assume to be relevant. In this study, we take a more open approach. Rather than starting from fully formed, predetermined frames, we investigate how ocean plastic research is framed in general, how these frames are constructed, and what elements they consist of. To do so, we analyse individual frame components, known as frame elements, in line with Entman's (1993: 52) widely cited definition of framing:

"To frame is to select some aspects of a perceived reality and make them more salient in a communicating text, in such a way as to promote a particular problem definition, causal interpretation, moral evaluation, and/or treatment recommendation for the item described."

Following Entman's definition of framing, the context of texts regarding ocean plastic research is determined by frame variables highlighting problems caused by ocean plastic, causes of problems, moral evaluations (who is responsible for causing and mitigating ocean plastic), and treatment recommendations (Vonk et al., 2024a). By considering individual frame-building elements rather than at entire frames, it becomes possible to understand which parts of frames are often replicated in newspaper articles and which parts are added, changed or omitted by journalists. This approach allows us to better understand the role of

press releases in frame construction in newspaper articles and shows what types of contextualization are preferred by journalists when communicating peer-reviewed ocean plastic research. In doing so, we hope to answer the following research question:

RQ1: How do frames change during the transfer from press release to newspaper article?

Understanding how ocean plastic research is framed in newspaper articles, and the role press releases play in shaping this framing, is important, as media communication directly influences how the public perceives and understands marine risks (Kramm et al., 2022). Scientific press releases often address both the environmental and human health impacts of ocean plastic pollution, though they tend to emphasize environmental concerns (Vonk et al., 2024). Similarly, newspaper articles more frequently present ocean plastic as a threat to marine ecosystems rather than to human health (Henderson & Green, 2020). However, studies show that environmental problems are perceived as more urgent and personally relevant when linked to human health rather than to distant ecological effects (Nisbet, 2009). Consequently, this ecosystem-focused framing may reinforce the perception of ocean plastic as a remote issue with little relevance to people's daily lives (Henderson & Green, 2020). How ocean science is framed in the media therefore shapes not only public understanding but also people's motivation to engage with and act on marine environmental problems (Caruso et al., 2022; Kelly et al., 2021).

#### 2.2 Narratives to communicate science

125

130

150

A narrative is a structured way of telling a story that connects events, characters, and emotions to convey meaning and shape how audiences understand a topic (Dahlstrom, 2014). In operationalizing narratives, we build on the approach of Vonk et al., (2024a), who adapted their narrative analysis from the four narrative dimensions identified by Glaser et al. (2009), including dramatization, emotionalisation, stylistic devices, and personalization. *Dramatization* concerns the structure of the story: whereas traditional factual reporting typically follows the inverted pyramid style, narratives often unfold with a clear beginning, middle, and end, creating tension and fostering engagement (Zebra, 2008). *Emotionalisation* refers to the expression of emotions, such as joy, anger, sadness, or surprise, by actors within a story, making their experiences and feelings more relatable to the audience (Glaser et al., 2009). *Stylistic devices*, including analogies and metaphors, are frequently used in science communication to make abstract or complex topics more accessible and to describe phenomena that extend beyond everyday human experience (Dahlstrom, 2021; Forgács & Pléh, 2022). Lastly, *Personalization* involves presenting abstract scientific issues in concrete human experiences by focusing on individuals or small groups and exploring their actions and the consequences thereof (Schiffer & Guerra, 2015).

Personalization helps bridge the gap between the reader and the subject matter, fostering emotional engagement and creating a sense of closeness with the material (Sangers et al., 2020). It allows audiences to identify with specific situations and feel empathy for the individuals involved (Dahlstrom, 2014). Characters within a narrative can be assigned distinct roles, such as victim, hero, or villain, which help the reader understand the type of characters that further shape the story context (Schwarze,

2006). Quotes subsequently serve as a means to convey the thoughts of key actors (Glaser et al., 2009). In science reporting, quotes increase the credibility and objectivity of an article and characterize the person quoted (Haapanen, 2017). As such, they help make scientific research more accessible to non-specialist audiences, allowing readers to hear the interpretive voice of the scientist directly (Hyland, 2010).

In scientific press releases on ocean plastic research, these narrative techniques primarily center on scientists, who explain important aspects of their research through quotes, positioning them as heroes or warners in the context of ocean plastic pollution (Vonk et al., 2024a). As a result, only the scientists' emotions are represented in the press releases, and the stories tend to personalize the scientific process itself, rather than the broader societal implications of ocean plastic pollution.

Journalists commonly adopt quotes from press releases directly (Autzen, 2014; Sharp et al., 2021), which can reinforce the scientist-centred narrative in newspaper articles and may cause the social dimensions of ocean problems to be underreported.

To examine if newspaper articles employ narrative elements differently compared to press releases, and to find out if journalists go beyond personalizing science by including non-scientific actors and actor quotes compared to press releases, we pose the following questions:

- RQ2: What are the differences in narrative elements such as dramatization, personalization, emotion, and stylistic devices between newspaper articles and the scientific press releases on which they are based?
- RQ3: Do newspaper articles contain different actor roles and quotes compared to the press releases on which they are based?

Part of the narrative is Story tone. Story tone, along with frame- and other narrative elements, can influence how audiences perceive and respond to information about environmental problems. In the context of ocean health, positive messaging has been shown to promote public engagement and support for environmental action (Kelly et al., 2022), as optimistic stories can inspire hope and highlight opportunities for recovery (McAfee et al., 2019). While press releases on ocean plastic research show a range of tones, negative, neutral, positive or passionate (Vonk et al., 2024a), press releases highlighting more negative aspects of pollution are more often followed up by newspaper articles (Vonk et al., 2024b). It is important to understand how negative messages about the ocean come about in public discourse, as scientists have raised concerns that a consistently pessimistic portrayal of ocean health in the media may discourage public engagement by reinforcing the perception that ocean degradation is irreversible (Duarte et al., 2015). It is well known that news selection criteria can create a bias towards negative news. Although science journalists report that tone is not a primary newsworthiness criterion in science news selection (Badenschier & Wormer, 2012), negativity remains a dominant news value in wider journalism (Bednarek & Caple, 2014). At the same time, positive news also plays a role in selection processes (Harcup & O'Neill, 2017), suggesting that both uplifting and alarming stories can attract media attention, depending on the context. While it is

known that tone can affect what is selected as news, less is known about how the tone of press releases affects the tone of subsequent coverage in newspapers, a question we aim to answer in this study.

This leads us to our fourth research question:

- RQ4: Does story tone differ between newspaper articles and the scientific press releases on which they are based?
- This theoretical framework combines framing elements, narrative techniques, actor roles, and story tone to provide a multidimensional lens through which we analyze how communication on ocean plastic research is transformed as it moves from scientific institutions to the public through newspaper coverage.

#### 3. Methods

#### 3.1 Dataset

- To analyse framing and narrative differences in ocean plastic science reporting, we compiled a dataset of 10 press releases and 130 subsequent newspaper articles about peer-reviewed ocean plastic research. All press releases were published on EurekAlert!, an international platform where research institutions share findings without editorial changes, making it possible to trace how the framing of research by research institutes translates into news articles worldwide. The press releases were published by research institutes in Canada, England and the United States. The 130 newspaper articles appeared in 75 newspapers across 13 countries, namely: Australia, Canada, China, India, Ireland, Kenya, New Zealand, Pakistan, Singapore, Thailand, England, and the United States. Newspapers ranged from international quality newspapers such as The Times and The Guardian to local media. Newspaper articles were identified using NexisUni and Altmetric, based on author names, paper titles, and affiliated institutions.
- This dataset is a subset of a larger collection compiled by Vonk et al., (2024b), which includes 84 EurekAlert! press releases published between January 2017 and December 2021, along with 495 related newspaper articles. These articles were published within one month of the corresponding press release, under the assumption that they were likely triggered by the publication of the peer-reviewed research, rather than by unrelated events that made the topic newsworthy. All newspaper articles were identified via NexisUni and Altmetric. By drawing from this pre-existing dataset, we focus on studies that received broad media coverage (i.e., at least nine newspaper articles based on a press release). To analyse how peer-reviewed studies are framed, only newspaper articles that discuss the research as their central topic are included; articles that cover multiple unrelated studies are excluded. The full selection criteria are provided in Appendix A, a visual overview of the selection process is added to the supplementary materials.

## 3.2 Coding

- To analyse framing and narratives in ocean plastic reporting, we used the codebook developed by Vonk et al. (2024a), which is grounded in Entman's (1993) widely cited definition of framing. This approach breaks frames down into four core elements: problem definition, causal attribution, moral evaluation, and treatment recommendation. Given that all press releases focused on scientific studies about ocean plastic, causal attribution was consistently coded as "ocean plastic." Problem definitions were categorised by the types of consequences described, including biological, economic, and social impacts, as well as implementation challenges or conflicts. Positive impacts of ocean plastic, such as the creation of new habitats, or positive consequences due to the working of mitigation strategies were also coded. Moral evaluation captured who was portrayed as responsible for causing or resolving the problem and whether the press release expressed a sense of urgency. Finally, treatment recommendations were identified when specific actions, solutions, or strategies were proposed to address ocean plastic pollution or its consequences.
- In addition to Entman's four frame elements, we extended the original codebook to assess localisation by introducing the variable local-specific information, which captures instances where the text refers to culturally, nationally, or geographically relevant details.

Narrative elements were coded following Glaser et al., (2009), focusing on dramatization, stylistic devices, emotionalization, and personalization. Dramatization was identified using an inverted pyramid structure, common in journalistic writing.

Stylistic devices were coded when analogies, figure of speech or metaphors were present in the text. Emotionalization was coded when the emotion of actors was expressed, for example fear, shock, surprise or hope. Personalization was analysed by coding if persons were present in a story, who caused events or were affected by events. In addition, we coded if these actors had specific roles such as victim, villain, hero, or warner. We also recorded whether these actors were quoted, and categorized quotes by source: the study's scientists, external experts, or non-scientific actors. Given that journalists often rely on press release are release for quotes (Autzen, 2014; Sharp et al., 2021), we also noted if newspaper articles used quotes from the press release and we then coded whether these quotes were copied verbatim or if they were copied with minimal edits.

All texts were coded by a single coder using NVivo (version R14.23.1). Results were exported to Excel for further analysis and visualised using Python (see Fig. 1–3). The selected text in the Excel was then qualitatively analysed to identify if the meaning of frame- and narrative variables was the same or differed between press release and newspaper article (see the supplementary materials for the codebook and Excel containing all results).

## 3.3 Coding rules and validation

235

Different coding rules were applied depending on the type of variable. For quotes, we coded each line that contained a quotation. For actor roles, framing variables, and narrative elements, we coded the entire paragraph in which the relevant

- content appeared. It did not matter which paragraph containing the variable was coded in the text, as long as the paragraph provided substantive information about it. For example, if a paragraph described a biological problem caused by ocean plastic, such as the degradation of coral reefs, that paragraph was coded for the "problem definition" variable. If the same frame element was repeated in multiple paragraphs within the same article or press release, it was only coded once. This approach was chosen due to the length and variation in the texts (some spanning up to two pages) and the large amount of different codes. Frequently recurring variables (like biological problems) appeared in nearly every paragraph, while others (like treatment recommendations) occurred only once. During codebook testing, coding every instance of frequent variables proved inefficient and reduced accuracy, as it shifted the focus from identifying unique elements to capturing repetition. This coding approach ensured consistency in identifying the meaning of frames and narratives. However, it did not account for differences in wording, as the same text fragments were not always selected.
- Intercoder reliability was established. To ensure that even infrequent variables were included, a manual sampling strategy was applied. This produced a subset of 35 press releases and newspaper articles (25% of the full dataset), in which each variable appeared in at least 20% of the texts. The frame variables conflict and economic problems were largely absent in the dataset, and could therefore not be validated. Intercoder reliability was assessed using Krippendorff's alpha, which is particularly suitable for evaluating agreement for rare categories (Krippendorff, 2011; Krippendorff, 2004). Intercoder reliability was assessed differently for actor quotes compared to frame and narrative variables. For the latter, agreement was determined based on whether both coders identified the same meaning of a variable, regardless of the paragraph in which it appeared. In contrast, for actor quotes, intercoder agreement was calculated based on whether both coders selected the exact same wording from the text. This more precise approach was necessary to evaluate whether quotes in newspaper articles were copied verbatim from press releases or with minimal edits.
- After resolving discrepancies through discussion, high intercoder reliability was achieved for nearly all variables. The exception was stylistic elements, which had a low agreement on *qualitative interpretation* ( $\kappa = 0.17$ ), though coders consistently identified their *presence* ( $\kappa = 0.87$ ). As a result, we refrain from making interpretations regarding differences in specific stylistic elements used. Frame variables had an average kappa score of 0.92 (ranging from  $\kappa = 0.84$  to  $\kappa = 1.00$ ). Narrative variables i.e., dramatization, emotion, and personalization, as well as actor roles had an average kappa score of 0.91 (ranging from  $\kappa = 0.76$  to  $\kappa = 1.00$ ). Quotes had an average kappa score of 0.89 (ranging from  $\kappa = 0.80$  to  $\kappa = 0.94$ ). Story tone had an average kappa score of 0.84 (ranging from  $\kappa = 0.76$  to  $\kappa = 0.91$ ). For a full overview of inter-coder reliability scores per variable, see Table 1A, Appendix B.

#### 4. Results

### 4.1 Framing in press release vs. newspaper article

To answer RQ1 and assess whether newspaper articles use different frame-building elements than press releases, all frame variables were visualized (Fig. 1). The results show considerable variation: 18 newspaper articles mirrored the press release in framing, these articles had a mean word count of 486. The majority of newspaper articles (N=70) were shorter (mean word count 328 vs. 689 in the press releases on which they were based) and contained fewer frame variables, often focusing solely on results. In contrast, 15 newspaper articles added frame elements (mean word count: 562), and 27 both added and omitted variables (mean word count: 615). These findings suggest that newspaper journalists typically reduce framing by shortening press releases, but sometimes reframe the content to provide additional context.

Figure 1: Visual representation of frame variable variation between press release (PR) and newspaper article (NA). The numbers of press releases and newspaper articles are provided at the top of the different cases. Newspaper IDs refer to entries in the dataset provided in the supplementary materials. Newspaper articles with red IDs have at least 20% less words than the PR, newspaper articles with cyan IDs have at least 20% more words than the PR. Each coloured line in the figure represents one frame variable, as explained in the legend to the right of the figure. The first block on the left shows the press release, with its internal framing indicated by the coloured lines. The blocks to the right represent the newspaper articles derived from that press release, whereby the colored lines indicate the same,

or different framing compared to the press release. This allows us to see which elements of the original framing are preserved, which are 285 frequently dropped, and which are added during the transfer from press release to news coverage.

# 4.1.1 Consequences of ocean plastic pollution

In both press releases and newspaper articles, ocean plastic was rarely framed as societal problem affecting daily live. Press releases mainly frame ocean plastic as **biological problem**, emphasizing problems to marine life, such as: animals ingesting plastic, getting entangled in plastic or coral reefs suffering from plastic-related diseases. These biological problems were repeated in newspaper articles when they matched research findings and were present in the press release, but were otherwise not emphasized. Press releases used precise scientific terminology, whereas newspapers simplified the language and presented the information more directly. For example:

- PR2: The scientists forecast that by 2025, plastic going into the marine environment will increase to roughly 15.7 billion plastic items on coral reefs, which could lead to skeletal eroding band disease, white syndromes and black band disease.
- NA 2.2: <u>Plastic waste is destroying coral reefs by spreading diseases</u>, a study has found.

**Non-biological problems** highlight the link between ocean plastic and broader environmental changes like climate change (e.g. changing weather patterns, warming ocean waters and ocean acidification). They were only discussed in press releases 2 and 3 and corresponding newspaper articles. Non-biological problems were never highlighted in newspaper articles when not present in the press release. **Problems related to treatment** appeared only in press release 7. Newspaper articles copied this framing when present in the press release, and otherwise almost never add it. Problems related to treatment differed between press release and newspaper article when newspaper articles quoted new societal actors that were not present in the press release, who provided additional criticism of ineffective solutions. For example:

- PR 7: The researchers note, however, that even if the prescribed effort is realized, the world remains locked into an unacceptable plastic future.
- NA 7.7: "The voluntary initiatives and commitments by the industry have failed," said Nusa Urbancic, campaigns director at the Changing Markets Foundation. "Policymakers should look past the industry smokescreen."

Opportunities related to ocean plastic were only mentioned in press release 3 and the corresponding newspaper articles. In this case, the study found that floating plastic debris created new habitats for coastal species that normally cannot survive in the open ocean. Newspaper articles replicated this positive framing by quoting the same researcher from the press release. Outside of this example, neither the press releases nor newspaper articles highlighted opportunities or positive aspects of plastic use. Opportunities resulting from mitigation mainly appeared in press release 1. Here, a researcher highlighted how reducing plastic waste could benefit coral reefs. Some newspaper articles copied this quote; others substituted a new quote from a

different researcher involved in the study. Subsequently, this substitute quote was widely repeated across newspaper articles. For example:

- PR 2: Said Lamb: "This study demonstrates that reductions in the amount of plastic waste entering the ocean will have direct benefits to coral reefs by reducing disease-associated mortality."
  - NA 2.7: Prof Bette Willis said: "Bleaching events are projected to increase in frequency and severity as ocean temperatures rise. There's more than 275 million people relying upon coral reefs for food, coastal protection, tourism income, and cultural significance. So moderating disease outbreak risks in the ocean will be vital for improving both human and ecosystem health."

# 320 4.1.2 Treatment recommendation and urgency

Press releases typically include **treatment recommendations** in the form of direct quotes from the scientists involved, often suggesting ways to reduce plastic pollution. Newspaper articles generally replicate this framing, either by copying these quotes verbatim or paraphrasing them. Some newspaper articles extend the framing in the press releases, by including suggested solutions from new actors:

• NA 10.9: "We can all make a difference by choosing not to buy fast fashion, which has a short shelf life in the shops and in our wardrobes, or by choosing to avoid plastic packaging and so on," he said.

Calls for **urgent action** to reduce the amounts of plastic or decrease our plastic use, are emphasized by scientists in press releases. Newspaper articles usually include such urgency only when it is present in the press release, but otherwise hardly add it. The call to action differs between newspaper articles and press releases when the newspaper article includes quotes from other (non-scientific) actors like environmental interest groups:

- PR 6: "Experts we surveyed found that entanglement in plastic and other pollution could pose a long term impact on the survival of some turtle populations and is a greater threat to them than oil spills. We need to cut the level of plastic waste and purse biodegradable alternatives if we are to tackle this grave threat to turtles' welfare."
- NA 6.8: Paul de Zylva, senior nature campaigner at Friends of the Earth, added: "It's time to get drastic with plastic. We can all help by re-using bags, carrying a reusable water bottle and cutting down on plastic packaging."

# 4.1.3 Responsibility for causing ocean plastic

Ocean plastic is rarely framed as problem **caused by individual plastic use** in either press releases or newspaper articles. Press releases never, and newspaper articles rarely, attribute responsibility for increasing amounts of plastic pollution to **political decision** making. In press releases, **countries** are held minimally responsible for plastic pollution, except in cases where the study explicitly focuses on this because the research tracks medical waste ending up in the ocean (PR 9). Newspaper articles more often highlight national responsibility, particularly when country-specific data is included, telling how much plastic a specific country used:

- NA 1.25: A 2015 Central Pollution Control Board survey found that 60 of India's major cities generated 15,000 tonnes of plastic waste every day. Of the waste that is collected an overwhelming amount of waste is neither recycled nor treated, it lands up in landfills or is dumped in water bodies.
- NA 2.15: Countries that take a great deal of care to keep plastic from entering the ocean -- like Australia -- see notably lower levels of it on reefs, and the problem was worst in those with poor infrastructure for managing waste, like Indonesia.

Both press releases and newspaper articles attribute responsibility to industry for causing ocean plastic pollution. In press releases, specific sectors are blamed, such as the fishing industry for discarding gear (e.g., PR 6 and PR 8), and hospitals for COVID-19-related plastic waste (PR 9). Newspaper articles often retain this framing. However, when journalists consult the original research article rather than relying solely on the press release, they may extend the framing of responsibility. For instance, PR 7 and PR 10 do not name specific industries, but the corresponding newspaper articles identify particular companies or sectors based on information from the original study that was not included in the press release.

• NA 7.7: One case study involved Coca-Cola Co., which Break Free From Plastic has repeatedly named the world's biggest plastic polluter..... The report also said that despite rhetoric to the contrary, beverage companies were actively undermining government efforts to curb plastic production by law.

#### 4.1.4 Responsible for mitigating ocean plastic

In press releases, scientists emphasize **society's** shared responsibility to reduce plastic pollution. These quotes consistently use the inclusive "we" form, implying that scientists themselves are part of the collective effort to address the issue:

• PR 6: We need to cut the level of plastic waste and purse biodegradable alternatives if we are to tackle this grave threat to turtles' welfare."

Newspaper articles seldom introduce society's responsibility for addressing ocean plastic pollution when this is not already mentioned in the press release. However, they frequently copy or paraphrase press release quotes that assign such responsibility to society. In some cases, journalists expand on this framing by including additional voices, such as environmental advocates or policymakers, and by providing local context through examples of regional initiatives or policies:

- NA 2.16: "<u>Individuals need to look at their own use of plastic</u>—bags, bottles, coffee cups," Mr. Silverwood said. "They feel so insignificant sometimes ... but it's a declaration of intent that <u>you don't want to be a contributor</u>."
- NA 1.30: "The 15-cent plastic bag levy was introduced back in 2002 to encourage Irish people to cut down on plastic waste."

Responsibility attributed to **countries** is limited in both press releases and newspaper articles. When present, responsibility is typically linked to political decision-making in specific countries or introduced through new quotes from scientists involved in the research, discussing possible treatments for ocean plastic:

- NA 1.30: <u>Ireland</u> has signed up to <u>aim to recycle 22 per cent of its plastic packaging waste</u> back into other plastic products.
- NA 9.1: To combat the influx of plastic waste into the oceans, the researchers urge for <u>better management of medical waste</u> in epicenters, especially in <u>developing countries</u>.

Several press releases (PR2, 7, and 4) stress the need for **policy** measures to mitigate plastic pollution. This framing is carried over to newspaper articles or expanded with additional quotes and references to national policies:

- PR 2: "While we can't stop the huge impact of global warming on coral health in the short term, this new work should drive policy toward reducing plastic pollution."
- NA 2.16: "We also believe it's vital the <u>NSW Government enacts a full ban to avoid facing an alarming future.</u>" A spokesman for Woolworths said it was on track to phase out bags at the checkout by July 1, as it announced last year.

The responsibility of industry for carrying out a treatment for plastic pollution is reinforced in newspaper articles, as this was absent in press releases. This reinforcement takes the form of new actor quotes or as references to local policies or initiatives aimed at decreasing plastic use:

• NA 6.10: "And if <u>CEOs of major companies like Coca-Cola</u> want to avoid being part of the problem then <u>they need to start</u> reducing their plastic footprint without delay."

#### 4.2 Narratives to communicate peer-reviewed research

To investigate how narrative elements are used to communicate peer-reviewed research on ocean plastic (RQ2), we analyzed story structure, personalization, stylistic elements, emotion, and story tone in both press releases and newspaper articles. All narrative elements are visualized in Fig. 2, which illustrates the variation in narrative use between press releases and newspaper articles.

Overall, most newspaper articles (N = 57) retained the same story tone as the corresponding press releases. Only one article shifted the tone to positive, 27 newspaper articles shifted to a negative tone, 13 articles shifted to a passionate or fatalistic tone and 19 articles changed the tone of the story to neutral. All press releases (N=10) and almost all newspaper articles (N=129) contain personalization. The inverted pyramid style is used in all press releases (N=10) and most newspaper articles (N=115). A few press releases (N=3) included emotional expressions from scientists. These emotions were reproduced in a limited number of newspaper articles (N=10). In a larger number of cases (N=39), newspaper articles introduced additional human emotion beyond what was present in the original press release.

Figure 2: Visual representation of the variation in narrative elements between press release (PR) and newspaper article (NA)

The numbers of press releases and newspaper articles are provided above of the different cases. For an detailed explanation of the figure,

please see the subtext of Fig. 1.

## 4.2.1 Dramatization, personalization, emotion and stylistic elements

Figure 2 shows that almost all press releases and newspaper articles have little dramatization, as they almost all use the inverted pyramid style, highlighting key findings in advance. Personalization is also common in both media, whereby scientists who conducted the research are positioned as main characters. Both press releases and newspaper articles use stylistic elements, especially metaphors, to make abstract or complex scientific concepts more accessible. Metaphors in press releases are copied in newspaper articles, or new stylistic devices, such as figurative language, are added:

- PR2: "What's troubling about coral disease is that once the coral tissue loss occurs, it's not coming back," said Lamb. "It's like getting gangrene on your foot and there is nothing you can do to stop it from affecting your whole body."
- NA 2.5: When coral reefs come in contact with plastic trash in the ocean, their risk of becoming diseased <u>skyrockets</u>, said an international study out Thursday.

Press releases convey emotion primarily through scientists' reactions to their findings. These emotional cues tend to be either positive: expressing excitement about new discoveries, or negative: emphasizing concern over the implications of plastic pollution. Newspaper articles frequently retain or even amplify this emotional framing, typically attributing it to the same scientists, as these are the main actors:

- NA 2.12: Lamb was surprised at how plastic is never mentioned as a threat to the aquatic ecosystem.
  - NA 10.1: Study leader Dr Ian Kane, from Manchester University, said: 'Almost everybody has heard of the infamous ocean
    "garbage patches" of floating plastic, but we were shocked at the high concentrations of microplastics we found in the deep-sea
    floor.

#### 4.2.2 Story tone

- Newspaper articles are coded more negative than the press release on which they are based, as these often leave out contextual information and focus on the problems caused by ocean plastic (RQ3). When press releases and newspaper articles use a fatalistic tone, doom language emphasizes the negative aspects of ocean plastic pollution or the 'enormous' amounts of plastic:
  - NA 5.4: Plastic <u>is destroying</u> life on the ground as well as underwater. We have stumbled upon <u>many heart-wrenching videos</u> of turtles who've consumed plastic.
- A passionate tone is present when scientists express enthusiasm about research findings; or when they tell that they believe they have found a treatment for problems related to ocean plastic:
  - NA 1.1: "It's <u>extremely, extremely exciting</u> because breaking down plastic has proved so challenging," said Paolo Bombelli from Cambridge University
- However, when such quotes are omitted, newspaper articles shift to a more neutral or hopeful tone. In these cases, stories highlight viable solutions, scientific advances or policy measures, creating a positive narrative that suggests that plastic in the ocean is a solvable problem.

# 4.3 Different types of actors in press releases and newspaper articles

To determine whether newspaper articles and press releases contextualize research differently by featuring different actors in specific roles, and by using different actor quotes (RQ3), we analyzed and plotted all actor roles and quotes in Fig. 3.

Figure 3: Visual representation of actor roles and quotes in press release (PR) and newspaper article (NA)

The numbers of press releases and newspaper articles are provided above of the different cases. For an detailed explanation of the figure,

please see the subtext of Fig. 1.

#### 4.3.1 Actor quotes

Nearly all press releases (n=9/10) include quotes from the scientists who conducted the study. Journalists copied these quotes either verbatim (n =44/121), or with minimal edits (n =16/121). Moreover, journalists added new quotes in newspaper articles from the scientists who conducted the study (n =50/121). Subsequently, these quotes were repeated across multiple newspaper articles. In contrast, few newspaper articles (n=14/130) featured quotes from non-scientists, such as politicians, corporate representatives, environmentalists, or citizens. An even smaller subset (n=9/130) featured quotes from scientists not involved in the study. When present, these quotes typically served to validate or contextualize the research rather than to introduce alternative perspectives:

• NA 2.13: "I'd never thought of bits of plastic as a vector of disease spread from the slime that coats them, but <u>the study shows</u> convincingly that corals entangled in plastic are 20 times more likely to be infected."

# 4.3.2 Actor roles

Press releases predominantly portray scientists in two roles: as heroes working to find solutions to plastic pollution, and as warners, warning society for the consequences of ocean plastic pollution. In contrast, newspaper articles depict a broader array of actor roles, depicting not only scientists but also but also communities, corporations, governments, and individuals as victims, heroes, warners, or villains.

Victims in newspaper articles include individuals or communities disproportionately affected by plastic pollution:

- NA 7.10: Across the globe, health problems associated with plastics production <u>disproportionately affect lower-income Black</u>,
   <u>Indigenous</u>, people of colour (BIPOC) communities
- NA 1.26: "We're designing chemicals and we're designing products because of their stability. We're a victim of our own success," says Dr. Reddy in a phone interview.

Villains include people littering, countries with poor waste management, high plastic discharge, or bad political decision-making; and corporate actors such as large companies producing plastic, like Coca-Cola and the petrochemical- and fishing industry:

- NA 1.15: Yet, even though people are aware of this concept, <u>many people are throwing waste in the Earth's waters.</u> In turn, this damages the aquatic ecosystem, especially coral reefs.
- NA 7.8: A case study involved <u>Coca-Cola Co.</u>, which Break Free From Plastic has repeatedly named as the <u>world's largest plastic polluter</u>.
- NA 2.12: Not much has changes in the recent years. A 2015 Central Pollution Control Board survey found that 60 of India's
  major cities generated 15,000 tonnes of plastic waste every day.

- Newspaper articles feature political actors or countries decreasing their plastic consumption as hero's. Also, companies trying to decrease their plastic use/production; international platforms or NGO's; newspapers and volunteers are featured in the hero role. However, in most cases, scientists finding or trying to find solutions to ocean plastic or related problems are the heroes of the story:
  - NA 8.6: The <u>Daily Mail</u> has <u>campaigned against plastic waste</u> for more than a decade
- NA 2.10: Separately, the <u>Queensland government has earmarked \$256 million</u> over the next five years to improve reef water quality
  - NA 2.16: <u>Harris Farm Markets has stopped offering single-use plastic bags</u> at the checkouts, the first major retailer to do so, and is encouraging customers to bring their own bag or take a recycled box from the store
  - NA 1.17: "We are planning to implement this finding into a viable way to get rid of plastic waste, working towards a solution to save our oceans, rivers, and all the environment from the unavoidable consequences of plastic accumulation."

In newspaper articles, warnings about plastic and its consequences are often issued by the scientists quoted in the press release. The newspaper then relies on their words by copying their quote or they add extra context by quoting environmental- or government agencies:

- NA 4.1: The growth of polluting microplastics in the Irish environment <u>has been confirmed by the Environmental Protection</u>

  Agency which has warned of the risk to public health.
- NA 7.3: <u>Ecologists from the University of Toronto are warning</u> of the monumental scale of action needed to curb plastic waste entering the world's rivers, lakes and oceans

#### 5. Conclusions and Discussion

We demonstrated that both research institutes and individual scientists can benefit from adopting a more strategic approach to crafting press releases, as these shape how scientific findings are framed in newspaper articles. Our analysis of press coverage on ocean plastic reveals that journalists often use quotes directly from press releases, meaning that scientists can influence the tone and narrative direction of resulting news articles. Moreover, research institutes play a key role in conveying a sense of urgency about environmental action, as this urgency is often carried over from press releases into news articles, but is rarely introduced by journalists when it is absent in the original press release. Rather than portraying ocean plastic as an isolated issue, press releases could strengthen ocean science communication by linking it to broader ocean challenges, emphasizing societal relevance, and clarifying responsibility. In this way, ocean science, often perceived as abstract or distant, can become more relevant to the public, and the need for action for a cleaner ocean is highlighted more clearly.

#### 5.1 The societal blind spot in ocean science communication

We found that newspaper articles build on press releases by adding contextual elements such as quotes from additional actors or region-specific examples. Similar to Sharp et al., (2021) these additions shift the framing of research in newspaper articles

compared to the original press releases. In our dataset, the most notable shift involves the attribution of responsibility: while press releases generally adopt a neutral or vague tone, newspaper articles more frequently refer to national policies or local governance, thereby placing the issue in a broader social context. However, these additions occur only to a limited extent. Most newspaper articles contain fewer framing elements than the original press releases and tend to focus primarily on the negative biological impacts of ocean plastic. This narrow framing may reinforce a sense of distance from the issue, as suggested by Henderson & Green (2020), who argue that portraying ocean plastic mainly as a biological problem can create the perception that it is not something people encounter in their everyday lives. As a result, ocean plastic remains an abstract concern for many, rather than a socially embedded issue requiring collective action.

As many people feel a distance from ocean science and plastic pollution (Schuldt et al., 2016), communication professionals, such as journalists and press officers, can help bridge this gap by using narrative storytelling and placing science in a relatable, human context (Dahlstrom, 2014). One key element of narrative construction is the use of direct quotes, which provide personal voices and emotional depth (Glaser et al., 2009). In some newspaper articles additional perspectives, such as those from policymakers, corporate actors, or NGOs, were included, thereby socially embedding the research. However, in most newspaper articles, the dominant narrative remained anchored on the voices of the scientists. This focus on science is also evident in previous research, which showed that press releases on ocean plastic often adopt a science-centered narrative and personalise the scientists involved, rather than placing the research in a broader social context (Vonk et al., 2024a). We find that this narrative is then replicated in newspaper articles based on these press releases.

The influence press releases have in the portrayal of ocean plastic research in the newspaper, highlights an important opportunity for science communicators to improve ocean science communication to broad audiences. For example, our analysis shows that broader connections, such as linking plastic pollution to climate change, only appear in newspaper articles when they are introduced in the press release, highlighting the press release's role in guiding the larger ocean narrative. These connections are important to make, as they can increase the relevance of topics like climate change to society (Pinto & Matias, 2023) and enhance public awareness of how the ocean functions (Kelly et al., 2022). Moreover, our analysis suggests that when press releases place research in a societal context, for instance by highlighting impacts on communities or assigning responsibility to industry and policymakers, journalists are more likely to adopt this framing in their coverage. This indicates that communication professionals can exert influence to achieve more socially informed journalism. In addition, journalists could increase the social relevance of plastic ocean pollution by actively engaging a wider range of voices. Doing so might help to shift the focus away from scientists and the personalization of science and instead make ocean problems more relatable to the public.

## 535 5.2 Journalistic use of sources in ocean plastic coverage

The reliance of journalists on press release content is concerning, as journalists, in their role as critical observers, are expected to independently verify and contextualize scientific claims (Göpfert, 2008). However, this watchdog function appears to be under pressure. Although newspaper articles typically contain fewer frame elements than press releases, their strong overlap in content underscores the influence of the press release on how scientific research is portrayed in the media (Vogler, 2020; Comfort et al., 2022). Moreover, journalists frequently reuse quotes from press releases, suggesting that these materials often serve as primary source material for newspaper articles (Nelissen & Hendrickx, 2023; Van Leuven et al., 2015). This pattern suggests a limited effort by journalists to seek external validation or include diverse perspectives, especially from scientists not directly involved in the research. One reason for this may be the difficulty journalists face in accessing relevant experts, an issue common in specialized fields like ocean science (Pinto & Matias, 2023). Additionally, when newspaper articles do include new quotes beyond those in the press release, these quotes often appear in multiple articles, indicating that they are likely sourced from wire services or international media rather than from original reporting (Nelissen & McMartin, 2022). This reliance on pre-packaged content reinforces the influence of research institutes, not only on what is reported, but also on whose voices are amplified in ocean plastic discourse.

#### 5.3 Ethical messaging and the role of research institutes in public discourse

In recent years, the communication departments of research institutions, including universities, have intensified their engagement with the media (Autzen, 2014). This shift is not solely aimed at disseminating scientific knowledge; in many cases, it also serves institutional goals such as building public reputation and visibility (Fürst et al., 2022). Hence, press releases are not only meant to communicate science, but also have clear public relation goals (Carver, 2014). When public relations objectives take the overhand, there is a risk that scientific nuance and caution are lost in favor of more appealing or sensational narratives. This dynamic has been well documented in health communication, where exaggerations found in press releases often translate directly into similarly exaggerated media coverage (e.g. Sumner et al., 2014, Bossema et al., 2019). This underscores the ethical responsibility of research institutions to ensure that their press releases present accurate, balanced, and contextualized representations of scientific findings. Given the strong influence press releases have on media coverage, as also shown in our results, research institutions should be seen as active agents in shaping public understanding of science. Their role goes beyond facilitating media uptake; it includes a responsibility to support truthful, nuanced, and socially responsible science communication.

The strong influence research institutes have on how science is portrayed in the media raises important ethical questions about responsible science communication, especially when findings call for urgent action or implicate societal actors, as in the case of plastic pollution. While some scientists warn that advocacy can compromise scientific neutrality (Büntgen, 2024), others argue that science is never neutral and that scientists have a moral duty to promote action on pressing issues (Van Eck et al.,

2024). Research also shows that the public, particularly in the context of climate change, expects scientists to not only inform, but also take a stance on policy matters (Cologna et al., 2021; Ulug et al., 2025). However, our data suggests that journalists rarely take such stances independently. In our dataset, attribution of responsibility, and calls for urgent action are typically included in newspaper articles only when they are present in the press release. For instance, although general media coverage of plastic pollution often avoids blaming industry (Schönbauer & Müller, 2021), articles in our dataset only framed industry as a polluter when the press release did so. Similarly, calls for urgent action were only replicated in news coverage if prompted by the press release.

# 5.4 Negative ocean plastic news

Overall, most newspaper articles either retained the tone of the corresponding press releases or adopted a more negative one.

Notably, around half of the press releases already conveyed a negative or alarmist tone, which contributed to newspaper coverage of ocean plastic often emphasizing the harmful and alarming aspects of pollution. The focus on negative aspects of ocean plastic is consistent with previous research showing that media coverage of plastic in the ocean often emphasises dangers and negative consequences (Schönbauer & Müller, 2021), while overlooking potential benefits or functional applications of plastic (Welzenbach-Vogel et al., 2022). The emphasis on ocean plastic problems in newspaper articles, reflects a broader media bias towards bad news (Hacup & O'Neil, 2017). However, this negativity bias was less pronounced in longer newspaper articles, which typically contained more framing elements and contextual information, resulting in a more balanced presentation of the issue.

Importantly, we found that the tone of newspaper articles was strongly influenced by the tone set in the press releases, particularly through the reuse of emotional quotes. In many cases, quotes expressing concern, optimism, or passion were directly copied from press releases, thereby shaping the story tone of the final newspaper article. Although the concepts of emotional narrative elements and tone are closely related, they refer to distinct aspects of a text. Emotional (negative) narrative elements specifically refer to the emotions expressed by actors within the story, for example, a scientist expressing frustration or concern. In contrast, the tone of the story refers to the overall emotional impression it creates for the reader, which can arise not only from the emotions of characters but also from how the issue is framed. We found for example, that a text that emphasizes environmental decline without directly quoting emotional reactions can still convey a negative tone.

Given that optimistic messages and positive storytelling can encourage public engagement and support for addressing marine environmental challenges (Kelly et al., 2022), our research seems to indicate that press releases represent an opportunity for setting a constructive tone in media discourse. Including quotes that convey passion for science or optimism about environmental solutions might help to foster a more positive narrative, especially as such quotes are frequently reused by journalists (Autzen, 2014; Sharp et al., 2021). Such strategies could shift the dominant framing away from portraying the ocean

as "beyond repair" (Duarte, 2015) and toward optimistic stories that inspire action and highlight the possibilities for restoration (McAfee et al., 2019). Nonetheless, it is important that this optimism remains grounded in realism (Cvitanovic & Hobday, 2018), as overly optimistic messages may undermine the perceived urgency and extent of environmental problems (Hornsey & Fielding, 2016).

#### 600 5.5 Limitations and further research

While our coding scheme is reliable, it only captures explicitly stated information, potentially underrepresenting implicit framing. We for instance coded if texts made an explicit reference to the responsibility of society for causing ocean plastic. Texts did refer to consumer goods contributing to ocean plastic. However, when these references were not explicitly linked to societal responsibility, this responsibility was not coded. We created our newspaper dataset by assessing which newspaper articles are based on the same research as discussed in press releases. As a result, we cannot be completely sure that the newspaper article is based on the press release or even that the journalist who wrote the newspaper article saw the press release. We recognize that other factors, such as prior knowledge or other press material, can also influence framing. The qualitative analysis of the type of framing and the large overlap in framing between press releases and newspaper articles and the copying of quotes from press releases, however, suggests that in most cases, the journalists did read the press release and used parts of it to write the newspaper article. Another limitation of our study concerns the composition of the dataset, which comes from different English-speaking countries with diverse cultural backgrounds. Practices in science communication may differ between the United States and the United Kingdom, for example. In our analysis, we focused on the content of the press releases and news articles, rather than linguistic style or cultural context. Although we did not observe remarkable cultural variation in the meaning of these elements, it is possible that differences exist at the linguistic level, such as tone, word choice or stylistic framing, we acknowledge that there may be variation based on regional or cultural norms. However, this variation was beyond the scope of this study. Future research could build on this work by focusing on differences in communication to better understand how ocean plastic research is communicated in different cultural contexts.

We examined how scientific press releases influence the framing of peer-reviewed research in newspaper articles using a combination of quantitative and qualitative text analysis. To support this, we developed a novel visualisation method that highlights differences in framing, narrative elements, actor roles, and citations. This approach enabled detailed case analysis, revealed broader trends, and improved transparency by linking findings directly to the data. By coding and visualising individual framing variables, we clearly identified shifts in framing between press releases and articles, and achieved high intercoder reliability even for complex elements like story tone. This method offers potential for automation, as coded variables can train machine learning models. Future research could build on this foundation by training models on larger datasets to systematically compare framing shifts across scientific disciplines, countries, or types of institutions, shedding light on broader patterns in science communication and media translation.

Our results show that journalists primarily modify framing related to social frame variables, such as attributions of responsibility. However, the social framing of science often remains narrow in newspaper articles on ocean plastic due to the limited diversity of actors and the underrepresentation of societal opportunities and problems. To better understand these framing choices, future research could conduct interviews with journalists and press officers. Such interviews may shed light on the challenges they face when reporting on ocean plastic research, the factors that influence their selection of quotes, and their rationale for including or excluding specific social problems and perspectives. These insights can help distinguish best practices in ocean science communication that can be used to bridge the gap between ocean science and society, ensuring that communication around environmental problems reflects both ecological urgency and social relevance.

# 635 Appendix A

To obtain a comprehensive understanding of potential modifications, we restricted our database to press releases covered by a minimum of 9 different newspaper articles. It is known that some news agencies copy content from other newspapers or that different newspapers belong to one news organization, causing the same article to be published multiple times in different newspapers. Hence, to avoid literal duplications in the dataset, we compared the content of all newspaper articles by calculating the Jaccard index in Python (See Appendix C for the Jaccard scores, the Python script is added to the online repository). All newspaper articles with a Jaccard index >0.8 were marked as duplicates, and only the first-published newspaper article was included in the analysis.

To analyse how a research institute frames research and to see if this framing differs from the framing in a newspaper article, we consider only newspaper articles that focus on the peer-reviewed research discussed in the press release, excluding newspaper articles that report on multiple studies (n=10). Moreover, press releases 3 and 45 and their newspaper coverage were excluded from the analysis since they covered the same scientific study, making data interpretation impossible as we would not be able to know with which press release we should compare the newspaper articles. To make it possible to evaluate trends in communication, we further excluded press releases with fewer than nine associated newspaper articles after all other exclusion criteria were applied. For a detailed explanation of the dataset construction and a visual representation, please refer to the flow chart in the supplementary materials.

#### Appendix B

All variables are coded separately and discussed in detail in the codebook (added to the supplementary materials) which contains several examples.

Tabel A1: Intecoder reliability results frames and narratives

| Variabele              |                                       | Kappa score | Presence in 2 <sup>nd</sup> -coder dataset |  |
|------------------------|---------------------------------------|-------------|--------------------------------------------|--|
| Narratives             |                                       |             |                                            |  |
| Inverted pyramid style |                                       | 0.77        | 27                                         |  |
| Personalization        |                                       | 1           | 34                                         |  |
| Stylistic element      |                                       | 0.17        | 61                                         |  |
| Actor roles            | Victim                                | 1           | 3                                          |  |
|                        | Villain                               | 0.84        | 4                                          |  |
|                        | Hero                                  | 0.90        | 18                                         |  |
|                        | Warner                                | 0.83        | 16                                         |  |
| Quotes                 | Quote studies Scientist               | 0.92        | 15                                         |  |
| ~                      | New Quote studies Scientist           | 0.80        | 72                                         |  |
|                        | Quote copied verbatim from PR         | 0.94        | 20                                         |  |
|                        | Quote copied minimal edits from PR    | 0.87        | 9                                          |  |
|                        | Quote other actor                     | 0.89        | 22                                         |  |
|                        | Quote Scientist not involved in study | 0.91        | 6                                          |  |
| Emotion                | Negative emotion                      | 0.88        | 13                                         |  |
|                        | Positive emotion                      | 1           | 5                                          |  |
|                        | Other emotion                         | 1           | 3                                          |  |
| Story Tone             | Fatalistic                            | 0.76        | 9                                          |  |
|                        | Negative                              | 0.88        | 17                                         |  |
|                        | Neutral                               | 0.82        | 3                                          |  |
|                        | Positive                              | 0.9         | 7                                          |  |
|                        | Passionate                            | 0.84        | 4                                          |  |
| Framing                |                                       | •           | •                                          |  |
| Problems               | Human Centered                        | 1           | 11                                         |  |
|                        | Biological                            | 0.86        | 27                                         |  |
|                        | Non-Biological                        | 0.93        | 10                                         |  |
|                        | Treatment                             | 0.73        | 14                                         |  |
| Opportunities          | Due to cause: i.e. ocean plastic      | 0.92        | 7                                          |  |
|                        | Due to treatment of ocean plastic     | 0.91        | 7                                          |  |
| Responsible for        | Politics or governments               | 1           | 4                                          |  |
| Cause                  | Companies or Industries               | 0.85        | 9                                          |  |
|                        | Regions or Countries                  | 0.84        | 4                                          |  |
|                        | Human activity, society               | 0.88        | 16                                         |  |
| Responsible for        |                                       | 0.94        | 12                                         |  |
| treatment              | Companies or Industries               | 0.90        | 8                                          |  |
|                        | Regions or Countries                  | 1           | 4                                          |  |
|                        | Society                               | 0.92        | 8                                          |  |
| Urgency to take action |                                       | 1           | 14                                         |  |
| Treatment recommend    |                                       | 0.91        | 22                                         |  |
| Local specific informa | ution                                 | 0.93        | 9                                          |  |

# Appendix C

To avoid duplicate newspaper articles in the dataset, the content of all newspaper articles was compared by calculating the Jaccard index with Python. For the Python script that was used, please see the repository.

| Article 1           | Article 2          | Article 1 | Article 2            | Article 1            | Article 2            |
|---------------------|--------------------|-----------|----------------------|----------------------|----------------------|
| NA 1.38             | NA 1.7             | NA 3.29   | NA 3.24              | NA 38.10             | NA 38.52             |
| NA 1.27             | NA 1.13            | NA 3.29   | NA 3.27              | NA 38.36             | NA 38.52             |
| NA 1.27             | NA 1.29            | NA 3.29   | NA 3.28              | NA 38.54             | NA 38.55             |
| NA 1.29             | NA 1.13            | NA 3.29   | NA 3.6               | NA 38.7              | NA 38.54             |
| NA 1.30             | NA 1.8             | NA 3.29   | NA 45.1              | NA 38.7              | NA 38.55             |
| NA 1.37             | NA 5.11            | NA 3.29   | NA 45.18             | NA 42.10             | NA 42.13             |
| NA 1.38             | NA 1.19            | NA 3.29   | NA 45.19             | NA 42.10             | NA 42.14             |
| NA 1.47             | NA 1.14            | NA 3.29   | NA 45.20             | NA 42.10             | NA 42.14             |
| NA 1.7              | NA 1.19            | NA 3.3    | NA 45.21             | NA 42.10             | NA 42.16             |
| NA 17.1             | NA 17.2            | NA 3.6    | NA 45.1              | NA 42.10             | NA 42.17             |
| NA 17.1             | NA 17.3            | NA 3.6    | NA 45.18             | NA 42.10             | NA 42.17             |
| NA 17.1<br>NA 17.2  | NA 17.3<br>NA 17.3 | NA 3.6    | NA 45.18<br>NA 45.19 | NA 42.10<br>NA 42.10 | NA 42.18<br>NA 42.19 |
| NA 17.2<br>NA 25.11 | NA 17.3<br>NA 25.6 | NA 3.6    | NA 45.19<br>NA 45.20 | NA 42.10<br>NA 42.13 | NA 42.19<br>NA 42.14 |
| NA 25.11            | NA 25.11           | NA 3.7    | NA 45.14             | NA 42.13             | NA 42.14             |
| NA 25.12            |                    | NA 3.8    | NA_45.14<br>NA_45.15 | NA 42.13             | NA 42.16             |
|                     | NA_25.6            |           |                      | _                    | _                    |
| NA_3.10             | NA_45.2            | NA_35.15  | NA_35.43             | NA_42.13             | NA_42.17             |
| NA_3.11             | NA_45.16           | NA_35.15  | NA_35.62             | NA_42.13             | NA_42.18             |
| NA_3.12             | NA_45.13           | NA_35.20  | NA_35.15             | NA_42.14             | NA_42.15             |
| NA_3.14             | NA_45.11           | NA_35.20  | NA_35.21             | NA_42.14             | NA_42.16             |
| NA_3.15             | NA_45.23           | NA_35.20  | NA_35.43             | NA_42.14             | NA_42.17             |
| NA_3.16             | NA_45.12           | NA_35.20  | NA_35.62             | NA_42.15             | NA_42.17             |
| NA_3.17             | NA_45.10           | NA_35.21  | NA_35.15             | NA_42.16             | NA_42.15             |
| NA_3.17             | NA_45.9            | NA_35.21  | NA_35.43             | NA_42.16             | NA_42.17             |
| NA_3.2              | NA_45.27           | NA_35.21  | NA_35.62             | NA_42.18             | NA_42.14             |
| NA 3.24             | NA 3.6             | NA 35.43  | NA 35.62             | NA 42.18             | NA 42.15             |
| NA 3.24             | NA 45.1            | NA 35.47  | NA 35.15             | NA 42.18             | NA 42.16             |
| NA_3.24             | NA_45.18           | NA_35.47  | NA_35.20             | NA_42.18             | NA_42.17             |
| NA_3.24             | NA_45.19           | NA_35.47  | NA_35.21             | NA_42.19             | NA_42.13             |
| NA_3.24             | NA_45.20           | NA_35.47  | NA_35.43             | NA_42.19             | NA_42.14             |
| NA_3.27             | NA_3.24            | NA_35.47  | NA_35.57             | NA_42.19             | NA_42.15             |
| NA_3.27             | NA_3.28            | NA_35.47  | NA_35.62             | NA_42.19             | NA_42.16             |
| NA_3.27             | NA_3.6             | NA_35.50  | NA_35.46             | NA_42.19             | NA_42.17             |
| NA_3.27             | NA_45.1            | NA_35.57  | NA_35.15             | NA_42.19             | NA_42.18             |
| NA_3.27             | NA_45.18           | NA_35.57  | NA_35.20             | NA_42.28             | NA_42.11             |
| NA_3.27             | NA_45.19           | NA_35.57  | NA_35.21             | NA_42.31             | NA_42.29             |
| NA_3.27             | NA_45.20           | NA_35.57  | NA_35.43             | NA_45.1              | NA_45.18             |
| NA_3.28             | NA_3.24            | NA_35.57  | NA_35.62             | NA_45.1              | NA_45.19             |
| NA 3.28             | NA 45.20           | NA 38.10  | NA 38.36             | NA 45.18             | NA 45.19             |
| NA 5.6              | NA 5.8             | NA 6.10   | NA 6.16              | NA 45.20             | NA 45.1              |
| NA 6.10             | NA 6.11            | NA 6.10   | NA 6.9               | NA 45.20             | NA 45.18             |
| NA_6.10             | NA_6.12            | NA_6.11   | NA_6.12              | NA_45.20             | NA_45.19             |
| NA_6.10             | NA_6.15            | NA_6.11   | NA_6.15              | NA_45.9              | NA_45.10             |
| NA_6.11             | NA_6.16            | NA_6.13   | NA_6.12              | NA_6.14              | NA_6.11              |
| NA_6.11             | NA_6.9             | NA_6.13   | NA_6.15              | NA_6.14              | NA_6.12              |
| NA_6.12             | NA_6.15            | NA_6.13   | NA_6.16              | NA_6.14              | NA_6.13              |
| NA_6.13             | NA_6.10            | NA_6.13   | NA_6.9               | NA_6.14              | NA_6.15              |
| NA_6.13             | NA_6.11            | NA_6.14   | NA_6.10              | NA_6.14              | NA_6.16              |
| NA_6.14             | NA_6.9             | NA_6.17   | NA_6.15              | NA_6.9               | NA_6.15              |
| NA_6.16             | NA_6.12            | NA_6.17   | NA_6.16              | NA_6.9               | NA_6.16              |
| NA_6.16             | NA_6.15            | NA_6.17   | NA_6.9               | NA_6.92              | NA_6.26              |
| NA_6.17             | NA_6.10            | NA_6.18   | NA_6.19              | NA_67.1              | NA_67.2              |
| NA_6.17             | NA_6.11            | NA_6.25   | NA_6.95              | NA_67.11             | NA_67.17             |

| NA_6.17  | NA_6.12  | NA_6.4   | NA_6.8   | NA_67.7  | NA_67.6  |
|----------|----------|----------|----------|----------|----------|
| NA_6.17  | NA_6.13  | NA_6.50  | NA_6.117 | NA_8.2   | NA_8.1   |
| NA_6.17  | NA_6.14  | NA_6.9   | NA_6.12  | NA_8.7   | NA_8.8   |
| NA_86.15 | NA_86.17 | NA_86.32 | NA_86.35 | NA_86.35 | NA_86.63 |
| NA_86.27 | NA_86.32 | NA_86.32 | NA_86.48 | NA_86.48 | NA_86.50 |
| NA_86.27 | NA_86.35 | NA_86.32 | NA_86.50 | NA_86.48 | NA_86.63 |
| NA_86.27 | NA_86.48 | NA_86.32 | NA_86.63 | NA_86.50 | NA_86.63 |
| NA_86.27 | NA_86.50 | NA_86.35 | NA_86.48 |          |          |
| NA_86.27 | NA_86.63 | NA_86.35 | NA_86.50 |          |          |

# Code availability

For the Python script that created all images and the script that calculated the Jaccard Index, see: <a href="https://github.com/erikvansebille/QualitativeDataVisualization/tree/main.">https://github.com/erikvansebille/QualitativeDataVisualization/tree/main.</a>

# Data availability

The codebook, dataset including all coded frame and narrative variables and the visual summary describing the dataset construction, as discussed in Appendix A, are all available as additional materials to this paper. All additional materials can be downloaded in Zenodo, ID: 10.5281/zenodo.15389206

#### **Author contribution**

AV, EvS and MB designed the study together. AV compiled the dataset, coded the data, analyzed the results and wrote the manuscript. EvS developed the Python script to visualize all results in Fig 1-3 and to calculate the Jaccard Index. MB second coded part of the dataset. SvB and MB both provided feedback on the manuscript, came up with ideas for the research and supervised during the research process.

#### **Competing interests**

The authors declare that they have no conflict of interest.

# 675 Ethical statement

As this research does not involve participants, the research did not need to be approved through the Utrecht University Ethical board.

## Acknowledgements

We would like to thank Niels Klaver for his role as second coder in this study. We used AI to check the manuscript for grammar and language mistakes.

# Financial support

No external funding for this project.

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
