# Peer review of "How frames and narratives in press releases shape newspaper science articles: the case of ocean plastic pollution."

_EGUsphere, 2025_

## Author Comment (AC1)

**Answers to review 1 of manuscript: How frames and narratives in press releases shape newspaper science articles: the case of ocean plastic pollution.**

Thank you for the opportunity to review this manuscript. You've done a great job demonstrating how the frames and narratives used in press releases can shape subsequent newspaper coverage. This study makes a timely and valuable contribution to the existing literature, and your writing is both clear and well grounded in prior research. To help clarify your findings, I've provided a few section-by-section suggestions below. I hope you find them useful as you refine your manuscript.

Thank you very much for your thoughtful and encouraging feedback. We truly appreciate your kind words about the clarity and contribution of the manuscript. Your detailed comments and section-by-section suggestions have been helpful in refining the paper, and in this response we address each of them. In our response, we have italicized all the information from the manuscript, the text that has been added is highlighted in red and we have added your feedback in blue.

**Abstract and Introduction**

The abstract clearly presents the main findings and their significance. The introduction clearly explains the different challenges that readers and journalists face when learning about ocean science.

1. In paragraph 1.1, split the challenges faced by the general public (e.g., need for prior knowledge, complex methods, physical distance) from those faced by journalists (e.g., limited scientist access, interpreting findings) into two separate paragraphs for greater clarity.

Thank you for this comment. The paragraph would indeed be more clearly structured if we addressed journalistic challenges and public challenges in understanding ocean science separately. We rewrote the text to two separate paragraphs. We have also added information on the journalistic challenges in communicating about ocean issues:

*Understanding the impacts ocean problems, like ocean plastic, can be challenging for people, as ocean problems are deeply interconnected and influenced by multiple stressors, making it difficult to isolate and fully understand the impact of a single issue (Kelly et al., 2022). Moreover, ocean science often requires the use of complex biological, chemical and physical methods. Understanding these methods can be complicated as they require a relatively large amount of prior knowledge to comprehend it properly. Additionally, the ocean, and particularly the deep sea, is largely invisible and physically remote, contributing to a sense of detachment and making it harder for the public to perceive its relevance to society (Schuldt et al., 2016). These communication challenges highlight the important role of journalists in making ocean science accessible to the public.*

*However, communicating about ocean science presents challenges for journalists too. In several European countries, journalists report that ocean science receives limited attention in newspapers, partly due to understaffed newsrooms and a shortage of specialized science reporters. Additionally, the complexity and technical nature of ocean science makes it difficult to accurately interpret research findings. Journalists often emphasize the need to consult directly with scientists to clarify and verify information, but such access is not always feasible. Time pressures further complicate their work, leaving journalists with little opportunity for in-depth investigation. As a result, they tend to rely more heavily on easily accessible international sources, often copying content related to ocean issues (Pinto & Matias, 2023).*

**Theoretical Framework**

You've defined the key concepts: framing, narrative use, overall tone, and actor roles, well.

1. In Section 2.1 you explain the concept of frame-building elements but don't discuss the effects frames can have on readers. (You do cover these effects in Sections 2.2 and 2.3 when discussing narratives

and story tone.) After defining frame-building elements in Section 2.1, include a brief overview of how frames influence readers' perceptions and interpretations.

Thank you for this valuable suggestion. To clarify the effects that frames can have on readers, alongside our definition of frame-building elements, we have added the following paragraph to the end of Section 2.1:

*Understanding how ocean plastic research is framed in newspaper articles, and the role press releases play in shaping this framing, is important, as media communication directly influences how the public perceives and understands marine risks (Kramm et al., 2022). Scientific press releases often address both the environmental and human health impacts of ocean plastic pollution, though they tend to emphasize environmental concerns (Vonk et al., 2024). Similarly, newspaper articles more frequently present ocean plastic as a threat to marine ecosystems rather than to human health (Henderson & Green, 2020). However, studies show that environmental issues are perceived as more urgent and personally relevant when linked to human health rather than to distant ecological effects (Nisbet, 2009). Consequently, this ecosystem-focused framing may reinforce the perception of ocean plastic as a remote issue with little relevance to people's daily lives (Henderson & Green, 2020). How ocean science is framed in the media therefore shapes not only public understanding but also people's motivation to engage with and act on marine environmental issues (Caruso et al., 2022; Kelly et al., 2021).*

**Methods**

The coding process is described clearly.

1. Please specify the timeframe during which the press releases and articles were published.

To clarify the timeframe of our dataset, we've added the following extra information to the second paragraph of 3.1:

*This dataset is a subset of a larger collection compiled by Vonk et al., (2024b), which includes 84 EurekAlert! press releases published between January 2017 and December 2021, along with 495 related newspaper articles. These articles were published within one month of the corresponding press release, under the assumption that they were likely triggered by the publication of the peer-reviewed research, rather than by unrelated events that made the topic newsworthy. All newspaper articles were identified via NexisUni and Altmetric. By drawing from this pre-existing dataset, we focus on studies that received broad media coverage (i.e., at least nine newspaper articles based on a press release). To analyse how peer-reviewed studies are framed, only newspaper articles that discuss the research as their central topic are included; articles that cover multiple unrelated studies are excluded. The full selection criteria are provided in Appendix A, a visual overview of the selection process is added to the supplementary materials.*

2. You note that inter-coder reliability for the stylistic elements did not reach an acceptable threshold. It may be cleaner to drop those elements from the analysis.

Thank you for this observation. It is correct that inter-coder reliability for the stylistic elements did not reach an acceptable threshold. While both coders identified a similar number of stylistic elements, they did not consistently identify the same instances. As a result, we do not draw any qualitative conclusions about the use or meaning of these elements in the Results or Discussion sections.

However, we prefer to keep this part of the analysis in the Methods and Results sections, as it reflects an important limitation of the codebook. By transparently reporting on the steps we took during our research and the limitations this caused, we want to make clear what conclusions can and cannot be drawn from our analysis and hope that others can learn from the limitations that emerged from our research.

**Results**

Figures 1–3 contain a lot of information. They're informative but require time to interpret.

1. You coded 38 elements (frame variables, narrative elements, actors, actor roles) but analyzed only 10 press releases. In the Results or Limitations section, discuss whether this sample size provides a sufficient basis for assessing all 38 coded elements.

Thank you for this important observation. In total, we analyzed framing across 140 data points (10 press releases and 130 newspaper articles), which provides a broader empirical basis for examining how these frame elements are transferred into news reporting.

This study builds on a previous analysis of framing in scientific press releases (see Vonk et al., 2024). Because our aim was to explore whether newspapers adopt similar framing as found in press releases, we used a comparable coding scheme. As a result, not every coded element appears in all ten press releases. For example, economic consequences are not discussed in the press releases included in our sample. However, what is particularly noteworthy is that such elements also do not appear in the corresponding newspaper articles. This suggests that journalists did not introduce them either and highlights the influence of the press release on the framing of news coverage.

2. In line 315, you refer to "press." I think you meant "press release."

Thank you for catching this. We have corrected the mistake.

3. For framing and for actors, you report the number of articles that mirror, add to, or omit elements compared to the press release. Please include similar counts for the narrative elements.

Thank you for your comment, we to make sure all results are in line, we have added counts for the narrative elements, like you suggested. The following information is added to the narratives' result section:

*Overall, most newspaper articles (N = 57) retained the same story tone as the corresponding press releases. Only one article shifted the tone to* positive, 27 newspaper articles shifted to a negative tone, 13 articles shifted to a passionate or fatalistic tone *and 19 articles changed the tone of the story to* neutral. *All press releases (N=10) and almost all newspaper articles (N=129) contain personalization. The inverted pyramid style is used in all press releases (N=10) and most newspaper articles (N=115). A few press releases (N = 3) included emotional expressions from scientists. These emotions were reproduced in a limited number of newspaper articles (N = 10). In a larger number of cases (N = 39), newspaper articles introduced additional human emotion beyond what was present in the original press release.*

**Conclusions and Discussion**

You link your results back to theory, suggest ways for journalists and scientists to improve communication, and identify the study's limitations and avenues for future research.

1. The structure of the Discussion does not mirror the order of the theoretical framework and Results. Reordering the Discussion subsections to follow frame-building, narratives, then actors would improve logical flow.

Thank you for this comment. You are correct that the structure of the Discussion does not mirror the order of the theoretical framework and Results sections. In an earlier version, the discussion was structured to match the manuscript; however, this led to substantial repetition, as the findings on frames, narratives, and actor roles are closely interrelated. For instance, the introduction of new actors or quotes often influenced the framing. To reduce redundancy and better highlight these interconnections, we chose to organize the discussion thematically rather than sequentially. We hope this approach improves coherence by emphasizing how these elements interact rather than treating them in isolation.

2. You mention article length in several places but do not integrate it fully into your findings. Either expand the analysis of length to show how it affects your results or remove those mentions if they do not support your main argument.

Thank you for this helpful feedback. Article length is indeed visualized in the figures, and for framing, we examined its influence more explicitly. In our results, we note that longer newspaper articles tended to include more framing elements, while shorter articles often omitted contextual details. This supports the idea that article length may influence the richness of framing.

For narrative elements, the influence of article length on the results is less evident. While article length is included in the figure presenting the narrative analysis, no clear pattern emerged. Framing is measured cumulatively, multiple frame variables add up, so longer articles often include more context and thus more frames, whereas shorter articles may omit these. In theory, this could also apply to narratives. However, since most articles follow the inverted pyramid structure and already include personalization by default, the cumulative effect is less visible in our results. Some articles include stylistic elements or emotional expressions from actors, but because these narrative indicators are limited in number, we observed no consistent link with article length. For this reason, we chose not to elaborate on article length in the narrative results or discussion.

Since we elaborate on text length only in the result section of framing, we have modified the discussion. Based on your previous feedback, we now mention that newspaper articles are often more negative than the press release they are based on (Results-3). Here, we do not address text length. We have changed the discussion as followed:

Text removed from the discussion: *In our dataset, shorter newspaper articles often focused only on the main research findings, without a broader context or discussion of societal implications. In our dataset, scientific findings were often linked to biological problems, causing the limited focus to contribute to a more negative overall tone in newspaper articles compared to press releases on which they are based.*

Text added to discussion*: Overall, most newspaper articles either retained the tone of the corresponding press releases or adopted a more negative one. Notably, around half of the press releases already conveyed a negative or alarmist tone, which contributed to newspaper coverage of ocean plastic often emphasizing the harmful and alarming aspects of pollution.*

3. You cannot be certain that press releases alone drive the framing of news articles, journalists may also draw on other sources. Please address this possibility in the Limitations section.

Thank you for this valuable observation. You are absolutely right that journalists may draw on a variety of sources beyond press releases when framing their news articles. We therefore have included the following information in the limitations section of the paper:

*We created our newspaper dataset by assessing which newspaper articles are based on the same research as discussed in press releases. As a result, we cannot be completely sure that the newspaper article is based on the press release or even that the journalist who wrote the newspaper article saw the press release. We recognize that other factors, such as prior knowledge or other press material, can also influence framing. The qualitative analysis of the type of framing and the large overlap in framing between press releases and newspaper articles and the copying of quotes from press releases, however, suggests that in most cases, the journalists did read the press release and used parts of it to write the newspaper article.*

---

## Author Comment (AC2)

**Reaction to Review 2 of manuscript: How frames and narratives in press releases shape newspaper science articles: the case of ocean plastic pollution.**

I think you have written a very interesting paper that fits nicely within the journal. I have had the great pleasure to review this and wish to stress that the comments are minor, meaning I hope to see this manuscript published soon. The paper is a unique and useful addition to the literature (as there is indeed limited knowledge on press releases (and their frames) and associated news articles). Below you'll find comments related to literature, structure and more general reflections.

Dear Miguel, thank you for your kind words on our manuscript and for taking the time to provide us with your valuable feedback. Your detailed comments and section-by-section suggestions have been helpful in refining the paper, and in this response we address each of them. In our response, we have italicized all the information from the manuscript, the text that has been added is highlighted in red and we have added your feedback in blue.

**INTRODUCTION**

I think you clearly sketch the relevance of the research and excite the reader to keep on reading.

Thank you for your kind words, we are happy to hear that you want to keep on reading due to the introduction.

Line 33-35: Maybe rephrase the sentence a bit. I struggled to understand the sentence due to it starting with 'as'.

You are right, the sentence is a bit hard to follow. We have rewritten it more clearly by taking the last sentence of the paragraph to the beginning:

*The media play an important role in shaping public understanding on ocean issues. They are a major source of information about marine science influence how risks, at least with regard to microplastics, are perceived among the public (Kramm et al., 2022).* *Although public awareness of ocean issues is increasing, perceptions of the most pressing threats often differ from those of scientists. For example, while the public often identify ocean pollution as their greatest concern, scientists are more likely to highlight climate change as the most urgent problem (Lotze et al., 2018). Scientists suggest that the disproportionate media coverage of ocean plastic, compared to other areas of ocean science (Pinto et al., 2020; Thompson-Saud et al., 2018), may have contributed to this gap in perception (Tiller et al., 2019).*

Line 45-50: Good relevance sketch. Could be combined with the last paragraph of the introduction (94-100).

Thank you for We have deleted the last paragraph of the introduction (lines 94-100) and added this information at the beginning of the paragraph (94-100). We have re-written the text as follows:

*In this study, we examine how scientific knowledge about ocean plastic spreads from research institutions to the public, focusing on the role of press releases in shaping media messages. Ocean plastic research forms a suitable case study because it is highly visible in both scientific communication and public discourse. While multiple studies show how ocean plastic is framed in public media, less attention has been given to the origins of these frames. Our study fills this gap, and explores contextual shifts among press releases and subsequent newspaper articles, by examining changes in framing and narrative strategies. By comparing how framing is used in both press release and newspaper article, we can better understand the role of the press release in frame construction. Furthermore, analysis of narratives, actors and quotes provides information about the social context in which ocean plastic research is placed and the people deemed important in the conversation about ocean plastic pollution.*

Line 52-59: Repetition of the word 'challenges' and 'challenging'

Based on the feedback of Anna and your feedback, we have re-written the paragraph:

*Understanding the impacts of ocean problems, like ocean plastic, can be challenging for people, as ocean problems are deeply interconnected and influenced by multiple stressors, making it difficult to isolate and fully understand the impact of a single issue (Kelly et al., 2022). Moreover, ocean science often requires the use of complex biological, chemical and physical methods. Understanding these methods can be complicated as they require a relatively large amount of prior knowledge to comprehend it properly. Additionally, the ocean, and particularly the deep sea, is largely invisible and physically remote, contributing to a sense of detachment and making it harder for the public to perceive its relevance to society (Schuldt et al., 2016). These communication challenges highlight the important role of journalists in making ocean science accessible to the public.*

Line 61: I'm not fully convinced of the importance of journalists based on your text yet. Maybe better to combine it with the next paragraph and truly stress the importance of these journalists.

Thank you for this observation, we have followed your advise and combined the information about the importance and the role of journalists in translating scientific information to the public with the following paragraph. We have re-written the text as follows:

*Effectively communicating ocean science to the general public requires translating complex research into content that is accessible and meaningful for newspaper readers. In doing so, journalists encounter a number of challenges. In several European countries, journalists report that ocean science receives limited attention in newspapers, partly due to understaffed newsrooms and a shortage of specialized science reporters. Additionally, the complexity and technical nature of ocean science makes it difficult to accurately interpret research findings. Journalists often emphasize the need to consult directly with scientists to clarify and verify information, but such access is not always feasible. Time pressures further complicate their work, leaving journalists with little opportunity for in-depth investigation. As a result, they tend to rely more heavily on easily accessible international sources, often copying content related to ocean issues (Pinto & Matias, 2023).*

Line 68-70: A bit off topic. Could be removed for me.

We agree with you, we also don't refer to this information in our discussion. Hence, we have removed the following section:

*Science journalism can additionally expose abuses such as unethical funding, plagiarism or methodological errors by taking a critical stance towards scientific claims and distinguishing between reliable and less reliable research (Fahy & Nisbet, 2011; 70 Lexchin, 2003; Murcott & Williams, 2013).*

Line 86: Maybe already explain the difference between narratives and frames a bit (not necessary however).

Thank you for this helpful suggestion. We have chosen not to elaborate on the difference between narratives and frames at this early point in the manuscript to maintain a clear and concise introduction. Instead, we explain both concepts in detail in the subsequent sections (2.1 and 2.2), where they are each introduced and contextualized more fully. We believe this placement avoids redundancy and allows for a more structured development of our theoretical framework.

**THEORETICAL FRAMEWORK**

Again, a strong outline of what is known and what you could add to the literature. Well done!

Thank you Miguel, for the positive evaluation of the theoretical framework. We are pleased to hear that you think we have provided a strong outline of the existing literature.

Line 107: The emphasis on 'how' threw me off a little bit. Framing always feels like a 'how', meaning I didn't truly understand the specific emphasis you wish to make or what the specific contrast is with the previous lines.

Thank you for your observant reading. The *how* should not have been in italic. We have written it without the emphasis.

Line 124-135: Very nice explanation of narratives. Just 'personalization' feels a bit out of balance due to the extra paragraph below (136-144). Maybe emphasize why personalization is big or combine 136-150. Also: be consistent with 'personalisation' or 'personalization'.

Thank you for this observation. We have revised the text to consistently use the spelling "personalization." We also understand that the emphasis on personalization previously felt unbalanced. To address this, we have merged the two paragraphs (lines 136–150) to ensure a more even and integrated discussion of narrative elements.

Line 155: 'And' between RQ's not necessary

We have deleted the 'And' between the RQ's.

Line 160: "How audiences perceive and respond to information". A question: Isn't this also the case with frames and narratives and not just the tone?

Thank you for this helpful comment. You are absolutely right, audiences' perceptions are shaped not only by tone but also by framing and narrative elements. We have revised the sentence on line 160 to clarify that all three elements, frames, narratives, and tone, play a role in influencing how readers interpret and respond to environmental information.

We have rewritten the line into: *"Story tone, along with frame- and narrative elements, can influence how audiences perceive and respond to information about environmental issues."*

Line 164: Definitely true that news has a 'negativity bias'. Nevertheless, Harcup and O'Neill also identify positive news. Maybe good to look at Caple & Bednarek (2016), as they are a bit closer to framing and discourse? Or Badenschier and Wormer (2012) for news values x science news? Just a tip!

Thank you for this valuable tip. You are right that these sources strengthen the theoretical background. Based on your feedback, we have revised this section to clarify the rationale behind our research question and to better integrate the relevant literature. We have re-written the paragraph as follows:

*Story tone, along with frame- and narrative elements, can influence how audiences perceive and respond to information about environmental issues. In the context of ocean health, positive messaging has been shown to promote public engagement and support for environmental action (Kelly et al., 2022), as optimistic stories can inspire hope and highlight opportunities for recovery (McAfee et al., 2019). While press releases on ocean plastic research show a range of tones, negative, neutral, positive or passionate (Vonk et al., 2024a), press releases highlighting more negative aspects of pollution are more often followed up by newspaper articles (Vonk et al., 2024b). It is important to understand how negative messages about the ocean come about in public discourse, as scientists have raised concerns that a consistently pessimistic portrayal of ocean health in the media may discourage public engagement by reinforcing the perception that ocean degradation is irreversible (Duarte et al., 2015). It is well known that news selection criteria can create a bias towards negative news. Although science journalists report that tone is not a primary newsworthiness criterion in science news selection (Badenschier & Wormer, 2012), negativity remains a dominant news value in wider journalism (Bednarek & Caple, 2014). At the same time, positive news also plays a role in selection processes (Harcup & O'Neill, 2017), suggesting that both uplifting and alarming stories can attract media attention, depending on the context. While it is known that tone can affect what is selected as news, less is known about*

*how the tone of press releases affects the tone of subsequent coverage in newspapers, a question we aim to answer in this study.*

Line 169-171: This explicit background regarding press releases is not given with other RQ's and is not necessary here, for me. You could leave it out.

Good to hear that it is also clear without the extra information, we have deleted the explanation.

**RESULTS**

I have close to no remarks on the result section, as I feel you have truly taken a unique approach (especially the figures) and have clearly explained what you found.

Thank you very much for your positive feedback on the results section. We greatly appreciate your kind words regarding the clarity of the findings and the use of figures and we are glad to hear that the approach came across as both clear and unique.

Line 337: COVID-19*-related plastic waste

Thank you for your observant reading, we have adjusted the text to: COVID-19-related plastic waste

Line 324-372: Just an idea! The result section reads a bit repetitive and maybe it would be good to combine the responsibility for causing and mitigating (as I was a bit confused on the first read), e.g. responsibility country cause x responsibility country mitigation. Just an idea, though.

Thank you for this suggestion. We understand that the section may come across as somewhat repetitive, especially because the concepts of *responsibility for causing* and *responsibility for mitigating* are closely related in name. However, we have deliberately chosen to present them separately, as they represent distinct frame variables with different implications. Combining them might obscure the nuance between attribution of blame and attribution of responsibility for action. For that reason, we have decided to keep the structure of the results section unchanged, to ensure conceptual clarity.

Line 384: Explicitly mention dramatization

Good idea, the inverted pyramid style is a measure for dramatization. Thank you for pointing this out, we have now incorporated dramatization in the explanation:

*Figure 2 shows that almost all press releases and newspaper articles have little dramatization, as they almost all use the inverted pyramid style, highlighting key findings in advance.*

Line 401-405: These lines are quite similar to the previous paragraph. How do you argue the difference between e.g. a negative tone and emotional (negative) narrative elements?

Thank you for pointing this out. We agree that the concepts of emotional narrative elements and tone are closely related, but they refer to distinct aspects of a text. We have clarified their distinction in the discussion section as follows:

*Importantly, we found that the tone of newspaper articles was strongly influenced by the tone set in the press releases, particularly through the reuse of emotional quotes. In many cases, quotes expressing concern, optimism, or passion were directly copied from press releases, thereby shaping the story tone of the final newspaper article. Although the concepts of emotional narrative elements and tone are closely related, they refer to distinct aspects of a text. Emotional (negative) narrative elements specifically refer to the emotions expressed by actors within the story, for example, a scientist expressing frustration or concern. In contrast, the tone of the story refers to the overall emotional impression it creates for the reader, which can arise not only*

*from the emotions of characters but also from how the issue is framed. We found for example, that a text that emphasizes environmental decline without directly quoting emotional reactions can still convey a negative tone.*

Line 437: Any idea why newspapers have a broader array of actor roles?

Thank you for this observation. One possible explanation for why newspapers include a broader array of actor roles is that they communicate scientific findings within the public sphere and aim to highlight their relevance to people's daily lives. By referencing local actors, political figures, or specific policy contexts, journalists can connect the research to broader societal debates and public concerns. In doing so, non-scientific actors become important narrative elements that help frame the story in terms of its social and political implications. While scientific press releases may point to the responsibilities of politics or broader governance, they tend to avoid naming specific political actors, probably because they focus on their own institution and keep the press releases broad in topic.

**CONCLUSION AND DISCUSSION**

You give a good conclusion and reflect on how your results fit within the literature in a proper manner. At certain parts I feel you could be a bit more normative (although you don't have to be) and expand on certain points.

Line 501-515: Could be used as a lead up to the 'ethical messaging' part. I feel the ethical messaging is a very important aspect of what you found (with the influence of press releases and the limited 'power' of journalists). I would therefore expand 529-540 a bit more. Focus on the responsibility of research institutes & their communication and perhaps add some more sources (e.g. Fürst et al, 2022) and see also Sumner et al. 2014 "our principle findings were that most of the inflation detected in our study did not occur de novo in the media but was already present in the text of the press releases produced by academics and their establishments".

Thank you for your suggestions and for pointing us toward relevant sources. You are right that the role of universities and research institutions in disseminating science news deserves more attention. The fact that press releases often serve public relations goals was underrepresented in our original discussion. Based on your feedback, we have expanded this part as follows:

*In recent years, the communication departments of research institutions, including universities, have intensified their engagement with the media (Autzen, 2014). This shift is not solely aimed at disseminating scientific knowledge; in many cases, it also serves institutional goals such as building public reputation and visibility (Fürst et al., 2022). Hence, press releases are not only meant to communicate science, but also have clear public relation goals (Carver, 2014). When public relations objectives take the overhand, there is a risk that scientific nuance and caution are lost in favor of more appealing or sensational narratives. This dynamic has been well documented in health communication, where exaggerations found in press releases often translate directly into similarly exaggerated media coverage (e.g. Sumner et al., 2014, Bossema et al., 2019). This underscores the ethical responsibility of research institutions to ensure that their press releases present accurate, balanced, and contextualized representations of scientific findings. Given the strong influence press releases have on media coverage, as also shown in our results, research institutions should be seen as active agents in shaping public understanding of science. Their role goes beyond facilitating media uptake; it includes a responsibility to support truthful, nuanced, and socially responsible science communication.*

APPENDIX

Line 595: Don't you mean you compared the content of all news articles, not press releases?

Thank you for your observant reading. We indeed mean newspaper articles. We have changed the text in the discussion accordingly.

Great work! Thank you for the opportunity to read and review this. And thank you for your valuable addition to the literature.

Thank you, Miguel, for your thorough reading and valuable feedback. Your suggestions have helped strengthen the manuscript. We also appreciate your kind words and are grateful for the opportunity to contribute to this area of research.

---

## Author Response (AR2)

Dear Louise Arnal,

Aike Vonk

We would like to thank you for handling our manuscript as editor and for your positive assessment of its quality.

We have added a table number and caption for the table under Appendix C. We have not adjusted the color scheme in Figure 3. The reason is that certain elements must share the same color. In the press release, blue indicates that a quotation from the scientist who conducted the study is present. When this quotation is copied verbatim into the newspaper article, the same blue color is used in the bar representing the news article, as the quotation is identical. If the quotation has been rephrased, or if a new quotation from the study's scientist is added, the bar is shown in light blue or dark blue, respectively.

We understand that this may appear counterintuitive at first glance, but we believe it is important to keep the color the same when the quotation in the press release and the newspaper article is identical. To clarify this point for readers, we have added the following explanation below Figure 3:

**Figure 3: Visual representation of actor roles and quotes in press release (PR) and newspaper article (NA)**

The numbers of press releases and newspaper articles are provided above of the different cases. Please note that the variables "Quote studies scientist" and "Quote copied verbatim" share the same colour. "Quote studies scientist" refers to a press release that includes a quotation from a scientist who conducted the study. When this quotation is copied verbatim into the newspaper article, it is coded as "Quote copied verbatim," hence the shared colour. If the quotation is rephrased or if a new quotation is added, these are shown in light blue and dark blue, respectively.

| We hope with this adjustment figure 3 is clear and the paper is ready for publication. |
|----------------------------------------------------------------------------------------|
|                                                                                        |
|                                                                                        |
| Kind regards,                                                                          |